# Application of ^1^H-NMR Metabolomics for the Discovery of Blood Plasma Biomarkers of a Mediterranean Diet

**DOI:** 10.3390/metabo9100201

**Published:** 2019-09-27

**Authors:** Shirin Macias, Joseph Kirma, Ali Yilmaz, Sarah E. Moore, Michelle C. McKinley, Pascal P. McKeown, Jayne V. Woodside, Stewart F. Graham, Brian D. Green

**Affiliations:** 1Institute for Global Food Security, School of Biological Sciences, Queen’s University Belfast, Belfast BT9 5DL, UK; smacias01@qub.ac.uk (S.M.); j.woodside@qub.ac.uk (J.V.W.); 2Beaumont Health, 3811 W. 13 Mile Road, Royal Oak, MI 48073, USA; jdkirma@gmail.com (J.K.); Ali.Yilmaz@beaumont.org (A.Y.); Stewart.Graham@beaumont.org (S.F.G.); 3Oakland University-William Beaumont School of Medicine, Rochester, MI 48309, USA; 4Centre for Public Health, Queen’s University Belfast, Belfast BT12 6BA, UK; sarah.moore@qub.ac.uk (S.E.M.); m.mckinley@qub.ac.uk (M.C.M.); 5School of Medicine, Dentistry and Biomedical Sciences, Queen’s University Belfast, Belfast BT9 7BL, UK

**Keywords:** biomarkers, dietary patterns, Mediterranean diet, metabolomics, ^1^H-NMR

## Abstract

The Mediterranean diet (MD) is a dietary pattern well-known for its benefits in disease prevention. Monitoring adherence to the MD could be improved by discovery of novel dietary biomarkers. The MEDiterranean Diet in Northern Ireland (MEDDINI) intervention study monitored the adherence of participants to the MD for up to 12 months. This investigation aimed to profile plasma metabolites, correlating each against the MD score of participants (n = 58). Based on an established 14-point scale MD score, subjects were classified into two groups (“low” and “high”). ^1^H-Nuclear Magnetic Resonance (^1^H-NMR) metabolomic analysis found that citric acid was the most significant metabolite (*p* = 5.99 × 10^−4^*; *q* = 0.03), differing between ‘low’ and ‘high’. Furthermore, five additional metabolites significantly differed (*p* < 0.05; *q* < 0.35) between the two groups. Discriminatory metabolites included: citric acid, pyruvic acid, betaine, mannose, acetic acid and *myo*-inositol. Additionally, the top five most influential metabolites in multivariate models were also citric acid, pyruvic acid, betaine, mannose and *myo*-inositol. Metabolites significantly correlated with the consumption of certain food types. For example, citric acid positively correlated fruit, fruit juice and vegetable constituents of the diet, and negatively correlated with sweet foods alone or when combined with carbonated drinks. Citric acid was the best performing biomarker and this was enhanced by paired ratio with pyruvic acid. The present study demonstrates the utility of metabolomic profiling for effectively assessing adherence to MD and the discovery of novel dietary biomarkers.

## 1. Introduction

The Mediterranean diet (MD) is regarded as a healthy dietary pattern with identified health benefits [1], including protective effects from cardiovascular disease (CVD) [2]. The MD is characterised by a high intake of fruits and vegetables, whole grains, fish, nuts, legumes, olive oil and moderate wine consumption [3]. It is also characterised by low consumption of dairy products, and highly processed foods such as sugar sweetened beverages, processed meats and sweets, fast food and trans-fat. In the PREDIMED (Prevención con Dieta Mediterránea) study, compared with a low-fat diet, an MD supplemented with nuts or virgin olive oil decreased the incidence of CVD [4]. Furthermore, when supplemented with nuts, it improved blood lipoprotein profiles resulting in a lower risk of atherosclerosis [5]. An inverse relationship between MD adherence and the development of metabolic syndrome has also been demonstrated [6]. A cross-sectional investigation of 8821 Eastern Europeans found significant associations between Mediterranean diet score (MDS) and waist circumference, systolic blood pressure and triglycerides [7]. Individuals with an MDS in the highest quartile were significantly less likely to have metabolic syndrome compared with those in the lowest quartile [7]. Similarly, an observational study of 3232 healthy French adults reported that adherence to MD was associated with a reduced risk of metabolic syndrome [8]. Furthermore, the PREDIMED study examined 808 participants at high risk of CVD, finding that subjects in the highest MDS quartile were at the lowest risk of metabolic syndrome, and were less likely to have elevated triglycerides or decreased HDL [9]. Adherence to dietary patterns is currently assessed by self-reported food frequency questionnaires and by 24h food recall. The limitations of these methods include lack of accuracy and potential for bias [10]. For example, fruit and vegetable intake is shown to be overestimated when participants record their own intake [11]. Several dietary scores have been developed and applied to populations to evaluate the role of diet in health. However, to overcome the disadvantages of traditional methods, it may be helpful if these practices are combined with novel biomarkers of food exposure. Combining these could more accurately measure adherence to a given diet [12]. Biomarkers of food intake are any component in the body fluids that can be utilised to indicate dietary exposure. When complemented with traditional dietary assessment practices, they may provide a more accurate picture of individual food exposure [11]. The lack of robust biomarkers of compliance is also a major challenge in clinical nutrition research, hindering the interpretation of clinical trials and observational studies of free populations. Hence, there is a need for reliable, non-biased, convenient and cost-effective methods to overcome this. In a Northern European population, MD adherence over a period of 12 months tended to increase plasma levels of vitamin C, oleic acid and eicosapentaenoic acid [13], but clearly, there is potential for the discovery of biomarkers associated with an MD pattern.

In this regard, metabolomics could be utilised for discovering robust dietary biomarkers [10]. Metabolomics facilitates the accurate measurement of many metabolites in body fluids, making it a useful tool for assessing the nutritional status of an individual [14]. Common metabolomic platforms are based on either ^1^H-Nuclear Magnetic Resonance (^1^H-NMR) spectroscopy or mass spectrometry. While the latter is more sensitive, ^1^H-NMR can provide robust, reproducible and unbiased results with very low pre-treatment requirements [15], making it a suitable platform for large cohort studies.

The majority of previously reported studies investigating adherence to the MD have focused on urine. ^1^H-NMR metabolomic profiling of human urine found that MD adherence was associated with changes in the levels of 3-hydroxybutyrate, citric acid, and *cis*-aconitate, oleic acid, suberic acid, various amino acids and some microbial cometabolites [1]. A second study, also involving ^1^H-NMR profiling of urine, identified 34 metabolites discriminating between low and high adherence to MD [12]. A third NMR study compared urine from individuals consuming an MD (supplemented with Coenzyme Q10) with those consuming a Western diet rich in saturated fat [16]. This found higher hippurate in the MD group, which positively correlated with Coenzyme Q10 and beta-carotene plasma levels. Finally, a fourth study undertook liquid chromatography-mass spectrometry (LC-MS) analysis of urine using a subset of participants from the PREDIMED study [17]. Specifically, this examined metabolomic differences between non-walnut eaters (who never consumed walnuts) and walnut eaters (> 3 servings/week). A total of 18 metabolites were higher in walnut consumers and these included fatty acid metabolites, polyphenol microbial metabolites and compounds related to the tryptophan/serotonin pathway. 

Thus far, only two studies have examined the blood metabolome of subjects consuming an MD. The first employed an LC-MS/MS methodology and indicated that the levels of several blood lipids are altered by an MD-based intervention [18]. The latter examined alternate MD correlation with serum metabolites and found that this diet was associated with 49 metabolites which positively correlated with different MD food components [19]. The overall aim of the current study was to discover novel blood-based metabolite biomarkers associated with the MD pattern. Therefore, we performed high-throughput ^1^H-NMR metabolomic analysis on blood plasma from a controlled study of MD adherence in a Northern European population. 

## 2. Results

### 2.1. Univariate Analysis

Following metabolomic profiling of 137 plasma samples from 58 subjects collected at three different time points (baseline, 6 months, 12 months), we were able to identify and quantify 59 metabolites in each of the ^1^H-NMR spectra. Citric acid was the most significant metabolite differing between Low and High MDS (*p* = 5.99 × 10^−4^*; *q* = 0.03). Moreover, five additional metabolites significantly differed (*p* < 0.05; *q* < 0.35) between the two groups. These were: pyruvic acid (*p* = 0.005), betaine (*p* = 0.014), mannose (*p* = 0.021), acetic acid (*p* = 0.030) and *myo*-inositol (*p* = 0.035). Citric acid and betaine were significantly higher in the high MDS group by 20.5% and 15.5%, respectively. Mannose, pyruvic acid and *myo*-inositol were 28.7%, 20.9% and 24.3% lower, respectively (Table 1). 

Table 1 shows the univariate statistical analysis of plasma metabolites in patients from Low (0–4) and High (5–10) MDS groups. Average concentrations (µM) of top 11 ranked metabolites and their respective standard deviations (SD). **p* < 0.05 (citric acid, *myo*-inositol, pyruvic acid, mannose and betaine) with concentration averages, false discovery rate (FDR), area under the curve (AUC) of the receiver operating characteristic (ROC) curve and % change.

### 2.2. Multivariate Analysis

Figure 1A shows principal component analysis (PCA) of the ^1^H-NMR data. Three potential outliers were highlighted, which prompted us to check the pharmacological profiles of these individuals. The profiles were broadly similar to the other participants and thus none were excluded. Principal component 1 (PC1) explained 29.5% of the variance and component 2 (PC2) explained 8.9% of the variance. Two supervised multivariate methods were then applied, orthogonal partial least squares discriminant analysis (OPLS-DA; Figure 1B) and partial least squares discriminant analysis (PLS-DA; Figure 1C), both of which improved the separation of groups (Figure 1B). The PLS-DA model was then cross-validated on Metaboanalyst using the 10-fold CV cross validation method and using a maximum of three components. (Appendix A), and by means of a Variable importance in projection (VIP) the top 15 ranking metabolites were identified. Citric acid, *myo*-inositol, pyruvic acid, mannose and betaine corresponded to the top five most discriminant metabolites in the model (Figure 1D). 

### 2.3. Correlation of Metabolite Concentrations with MDS, and Consumption of Food Components

Citric acid, pyruvic acid, betaine, mannose and *myo*-inositol data were non-normally distributed and metabolite correlations were examined using Spearman’s rank correlation coefficient. These five metabolites significantly correlated with the 14-point scale MDS. Citric acid and betaine positively correlated with MDS while mannose, pyruvic acid and *myo*-inositol negatively correlated (Appendix A) (Figure 2A). Application of multiple comparison using the Benjamini-Hochberg approach showed FDR to be higher than 0.05 (Appendix A).

Correlations were examined between each of the top five metabolites and each food group (Appendix A). These were: Fruit (Fr); Fruit Juice (J); Combined Fruit and Fruit Juice (FJ); Combined Fruit, Fruit Juice and Vegetables (FJV); Red meat (RM); combined Chicken and Turkey (CT); Fish (F); Nuts (N); Processed meat (PM), Legumes (L) and Alcohol (A); Olive Spreads (OS); Olive/Rapeseed oil (OR); Sweet Foods (S); Combined sweet foods and carbonated drinks (SD) and whole grain cereals (C). Citric acid positively correlated with J, FJ and FJV and negatively correlated with S and SD. Mannose significantly negatively correlated with J, FJ, FJV and C. Pyruvic acid significantly negatively correlated with J, FJ, FJV, F and OS and significantly positively correlated with PM. *Myo*-inositol was significantly negatively correlated with OS and positively correlated with PM, betaine significantly positively correlated with J and FJV. Application of multiple comparison using the Benjamini-Hochberg approach showed FDR to be higher than 0.05 (Appendix A). T-test conducted to evaluate smoker and non-smoker differences showed no metabolites significantly differed between smokers and non-smokers. The top five shortlisted metabolites were citric acid (*p* = 0.07), betaine (*p* = 0.18), mannose (*p* = 0.34), pyruvic acid (*p* = 0.34) and *myo*-inositol (*p* = 0.34). Food group-to-Food group correlations within the study cohort and correlations of all 59 metabolites against each food group scored within the MDS can be seen on Appendix A (Appendix A respectively)

### 2.4. Identification of Biomarkers

Table 1 shows the individual area under the curve (AUC) of the receiver operating characteristic (ROC) curve for each computed metabolite. Citric acid performed moderately well with an AUROC curve=0.67 which was superior to the other metabolites: pyruvic acid (0.64), betaine (0.62), mannose (0.61) and *myo*-inositol (0.60). Computed metabolite ratios showed improved ROC values. The citric acid/pyruvic acid was highest (0.74) (Figure 2C) followed by citric acid/lactic acid (0.73), citric acid/aspartate (0.72), citric acid/l-phenylalanine (0.71) and betaine/pyruvic acid (0.70) (Table 2).

Table 2 shows the best performing paired metabolite ratios with area under the curve (AUC) of the receiver operating characteristic (ROC) curve and % change. Features displayed in the table are ranked based on area under ROC curve with highest AUC (ROC) corresponding to the ratio Citric acid/Pyruvic acid, Citric acid/L-Lactic acid or Citric acid/Aspartate.

## 3. Discussion

Combining traditional methods such as food diaries or 24-h recall with biomarkers of food exposure may provide a more accurate picture for assessing adherence to a given diet. The present study used food diaries from 58 participants and quantified 59 blood metabolites by ^1^H-NMR and examined how their concentrations differed according to the reported Mediterranean diet score (MDS) of participants. Scoring systems are useful for measuring dietary patterns in a population but there is a need for continual improvement [20]. A previously validated 14-point item scale was adapted and adopted [3]. It was also employed in a UK population, demonstrating satisfactory accuracy for assessing MD adherence among individuals [21].

Adherence to the MD has often been assessed in Mediterranean populations where the MD patterns are regarded as familiar eating habits. Studying the efficacy of the MD in non-MD populations remains a challenge [22]. MD adherence in non-Mediterranean populations is generally lower [13]. In the MEDDINI cohort no participants achieved the highest MDS. The maximum MDS achieved in this non-Mediterranean population was 10, and “low” (0–4) and “high” (5–10) MDS groups were selected by dividing groups at the median. Unsupervised multivariate models incorporating all 59 metabolites only weakly separated these groups. This was modestly improved by the application of supervised modelling methods such as OPLS-DA and PLS-DA; however, it is clear that a degree of overlap exists between groups when the overall metabolomic profile is considered. This result is not surprising given the diverse range of components contributing to the MDS scoring system and the inherent heterogeneous character of the MD. Despite this, the multivariate statistical analysis identified five individual metabolites within the model, which were important for discriminating between “low” and “high” MDS groups. These were: citric acid, mannose, pyruvic acid, *myo*-inositol and betaine; therefore, we proceeded with univariate statistical analysis. 

These same five metabolites significantly differed between “low” and “high” MDS groups. Citric acid and betaine were significantly higher in the high MDS group, by 20.5% and 15.5%, respectively. *Myo*-inositol, pyruvic acid and mannose were lower in the high MDS group, by 24.3%, 20.9% and 28.7%, respectively. Area-under-the-curve (AUC) values of the receiver operating characteristic (ROC) curves were compared for every single individual metabolite and all paired combinations of metabolites. Citric acid was the most influential metabolite in discriminating between ‘low and ‘high’ MD groups. As a single biomarker, citric acid was the best performing, it was the most capable of discriminating between “low” and “high” MDS and achieved the highest AUC

ROC curve (0.67). Promisingly, this was greatly enhanced through combination with pyruvic acid. A metabolite ratio of citric acid and pyruvic acid achieved an AUC ROC of 0.74. 

In terms of the dietary components, citric acid positively correlated with the consumption of fruit juice, combined fruit and fruit juice and combined fruit, fruit juice and vegetables. It negatively correlated with the consumption of sweet foods and combined sweet foods and carbonated drinks. The strength of these relationships could perhaps be tested in larger cohorts with greater statistical power [23]; however, this is not the first time that citric acid has been associated with the MD. There is certainly evidence elsewhere to support a dietary link. Urine levels of citric acid were 26% higher in men and 35% higher in women in consuming a lactovegetarian diet compared with omnivorous control groups [23]. Similarly, in the PREDIMED study citric acid levels were significantly higher in urine from participants following an MD for 1 year compared with those following a low-fat diet [1]. This suggests that citric acid has a degree of specificity for the MD. There is evidence to suggest that where extra-virgin olive oil is consumed it could be a key source of dietary citric acid [1] and this specific association needs to be further explored. In contrast, there are metabolites reported in this study which were not significant in the present study, namely 3-hydroxybutyrate and *cis*-aconitate (these two, like citrate, were metabolites from the metabolism of carbohydrates, creatinine, amino acids (proline, *N*-acetylglutamine, glycine, branched-chain amino acids, and derived metabolites), lipids (oleic and suberic acids) and microbial cometabolites (phenylacetylglutamine and p-cresol)).

Other studies where adherence to MD was assessed explored the associations of an MDS with similar food groups. The alternate Mediterranean diet study found associations between MDS and fruit, vegetables, whole grains, fish and unsaturated fat. However, in this case, the highest point scale to assess adherence to MD was 8 and the highest score achieved was 4. This study found 21 identifiable metabolites associated with MD including four amino acids, one carbohydrate, two co-factors or vitamins, 11 lipids, and three xenobiotics [19].

Citric acid has previously been associated with resistant hypertension [24]. Our data showed a significant positive correlation between citric acid and both systolic and diastolic blood pressure. However, citric acid also proved to be significantly positively correlated to MDS. Correlations showed a stronger association and higher level of significance between citric acid and MDS (*r* = 0.280; *p*-value: 0.001) than those with blood pressure (*r* = 0.187; *p*-value: 0.032; *r* = 217; *p*-value: 0.012 for systolic and diastolic pressure, respectively). Systolic blood pressure did not significantly change throughout the intervention when doing an analysis of variance (ANOVA) comparing systolic blood pressure at baseline, 6 months and 12 months (*p*-value: 0.365.). Diastolic blood pressure showed significant differences between baseline and 12 months (*p*-value: 4.7 × 10^−4^) but not between baseline and 6 months (*p*-value: 0.138). The highest increase in citric acid was observed between baseline and 6 months. Citric acid showed a decrease between 6 and 12 months. Similarly, MD adherence was higher between baseline and 6 and decreased between 6 and 12 months, which strengthens a potential relationship between citric acid and MD adherence.

Pyruvic acid and citric acid are closely related metabolites in cellular metabolism and the results for pyruvic acid in the present study contrasted greatly with citric acid. Blood pyruvic acid was approximately 20% lower in individuals with a high MDS. Furthermore, pyruvic acid negatively correlated with the consumption of fruit, combined fruit and fruit juice, fish and olive spreads and positively correlated with processed meat. The negative relationship between pyruvic acid and fish consumption was the most statistically significant and strongest metabolite-food correlation observed. Furthermore, the positive correlation between pyruvic acid and processed meat was the second most statistically significant. It is worth remembering that all patients participating in the present study had a history of CVD or unstable angina, and that there was a significant difference in blood pressure between the two groups. Altered pyruvic acid has previously been associated with hypertension [25]. However, a correlation did not show any association between blood pressure and pyruvic acid in the present study. Excessive levels of pyruvic acid have also been linked with cancer [26], and even neurodegenerative diseases [27,28]. For example, pyruvic acid levels are around 20 times higher in highly invasive cancer cells compared with low invasive cancer cells [26], and pyruvic acid is two to four fold higher in diabetic rats than non-diabetic rats [29]. Pyruvic acid has not been previously linked to MD and further work should be completed to further elucidate this association. However, it is noteworthy that normal weight human subjects have significantly higher levels of both pyruvic acid and citric acid compared with severely obese patients [30]. In our dataset, patients’ weight was significantly positively associated with pyruvic acid but not citric acid (data not shown).

A number of other metabolite-food relationships were observed. Mannose significantly negatively correlated with fruit, combined fruit and fruit juice, combined fruit, fruit juice and vegetables and with whole grain cereals. We are not aware of any dietary-based studies that have been linked with mannose. However, the analysis of a large cohort of 2204 females found that after glucose, plasma mannose had the strongest metabolite association with type 2 diabetes [31]. 

We found that *myo*-inositol (the most abundant stereoisomer of inositol) negatively correlated with olive spreads and positively correlated with processed meat consumption. Neither fruit or vegetable consumption, nor red meat consumption were correlated with plasma *myo-*inositol levels. Initially, these observations seemed counter-intuitive given that *myo-*inositol occurs very abundantly in numerous plant-based foods associated with the MD, such as fruits, beans, grains and nuts [32]. However, is it well known that animal sources of inositol are more bioavailable than the phytate form, which is present in plant sources [33]. Furthermore, some of the richest, most bioavailable sources of inositol are organ meats, which may possibly explain the positive correlation observed for processed meat, but not for red meat. 

Betaine (a derivative of choline) is naturally present in a variety of plant based foods such as cereals, grains and leafy vegetables and beets [34,35] and has previously been linked to MD [36]. It also may be a useful biomarker for identifying participants following the DASH (dietary approach to stop hypertension) diet [37]. Our investigation found that betaine was 15.5% higher in the “high” MD group, and it positively associated with the consumption of fruit juice and combined fruit, fruit juice and vegetables. This would indicate that diet is an important contributor. Such a link remains controversial. A cohort case study from the PREDIMED trial involving individuals with high CVD risk divided into an MD group and a control group found no significant associations between individual metabolites from the choline pathway and CVD after one year. However, the ratio betaine/choline showed an inverse association with CVD incidence although not with stroke alone [38]. Furthermore, in a cross-sectional subset of the Nutrition, Aging, and Memory in Elders cohort, betaine was also associated with lower levels of low-density lipoprotein cholesterol and triglycerides and reduced risk of diabetes mellitus [39]. However, further studies in different populations are necessary to confirm the link between betaine and CVD.

Citric acid and betaine significantly and positively correlated to MDS (r = 0.28, r = 0.19 respectively). Mannose, pyruvic acid and *myo*-inositol significantly and negatively correlated to MDS (r = −0.2, r = −0.02, r = −0.18 respectively). Citric acid showed to be the highest and most significant correlation with MDS, however overall these associations showed to be weak and when applying multiple comparison using the Benjamini-Hochberg approach, the FDR did not show to be less than 0.05 (Appendix A). Further studies should be carried out to prove their association with MDS.

The limitations of the present study should be taken into consideration for future studies. A total of 58 patients participated in the study followed an MD during 12 months and plasma samples were collected at baseline, 6 months and 12 months. Samples were divided into two groups (low and high MDS) according to how well patients adhere to the 14-point scale MDS. Patients’ MDS was the lowest at baseline and it improved throughout the intervention; therefore, the majority of samples in the Low MDS group were from the baseline and the majority of samples in the High MDS group were collected at 6 and 12 months of the intervention. Despite the degree of dependency of the samples in both groups, this was not detrimental to the results. A parallel analysis was carried out dividing samples into timepoints: baseline, 6 and 12 months (paired samples) and results showed the same top five metabolites as the most significant ones: citric acid, betaine, mannose, *myo*-inositol and pyruvic acid. It is also worth noting that the concentrations of these five metabolites were not altered when applying paired or unpaired analysis.

Patients had a history of myocardial infarction or unstable angina previous to the intervention and were on a range of medication. However, drug intake diaries were revised to confirm that there was no association between outliers in the PCA model and drug intake. Patients were mostly overweight or obese; however, body mass index (BMI) did not show significant differences when comparing the low and high MDS groups. The present study focused on the discovery of biomarkers purely associated to diet and the effect of exercise was not recorded or taken into consideration. Blood pressure showed to be significant between the low and the high MDS groups, there were similar percentages of smokers in the low and high MDS groups and the percentage of males compared to females was higher in both groups. Gender did not show any significant differences in the results. Smoking status of individuals did not show any detectable influence on metabolite concentrations, although the *p*-value for citric acid was close to statistical significance. Further studies in larger cohorts could examine whether a relationship exists here. It should also be noted that none of the participants achieved the highest MDS on the 14-point scale and the highest scored achieved was 10. 

The present study conducted untargeted metabolomics with the goal of profiling the plasma metabolome of two groups with high and low adherence to MD. ^1^H-NMR was used due to its high precision, reproducibility and quantitative power. This study found significant direct associations between MD and citric acid and betaine and significant inverse associations between MD and pyruvic acid, mannose and *myo*-inositol. These metabolites were significant following univariate analysis, but multivariate analysis corroborated these by indicating that these five metabolites were most responsible for the discrimination of the model (VIP scores of the PLS-DA model). Out of the five metabolites, citric acid and betaine have also been associated with MD on previous studies. Nevertheless, following further studies, targeted metabolomics should be carried out to confirm the validity of these five metabolites and their association with MD. 

## 4. Materials and Methods

### 4.1. Patient Cohort 

The Mediterranean Diet in Northern Ireland *(*MEDDINI*)* study was a pilot randomised controlled parallel group trial where 61 willing participants previously diagnosed with coronary heart disease (CHD) were recruited (samples from n = 58 participants were available for the present study) from the Cardiology Directorate, Royal Victoria Hospital, Belfast. Patients recruited were aged between 39 and 78 years and had a recent diagnosis (within four weeks prior to starting the study) of myocardial infarction (MI) or unstable angina. Patients with severe heart failure, pending on immediate inpatient coronary revascularisation, on warfarin therapy, on Omacor therapy, with cognitive impairment, those with records of extreme alcohol intake, taking multivitamin/fish oil supplements, unable to comply with the diet or to provide informed consent or those who were not expected to live longer than 6 months for any other causes were excluded from the study [13].

Plasma samples and dietary assessment measurements were collected at three-time points; baseline, 6 months and 12 months. Seven-day food diaries were used to collect food consumption data. Participants were asked to record the foods consumed over seven consecutive days, including an estimation of quantity consumed and information on preparation methods used. From seven-day food diaries, a database was created registering all food amounts eaten by all participants during the course of the intervention (baseline, 6 months and 12 months). 17 selected food groups from this database were used to correlate with shortlisted biomarkers (Appendix A). Fasting blood samples were collected at the three time points to assess nutritional biomarkers. Ethical approval was obtained for this study from the Queen’s University Belfast Research Ethics Committee (ethical approval references: RGHT000049 and 15.42 for the Meddini original study and for the latest analysis respectively. Informed written consent was obtained from all participants [13].

The baseline characteristics of the sampled cohort (average BMI, blood pressure, age, sex and smoking characteristics) can be found in the Appendix A. Mean age was 56.1 years for the low MDS group and 60.5 y for the high MDS group, and participants were on a range of medications. Mean BMI was 30.2 kg/m^2^ for the low MDS group and 29.8 for the high MDS group, which indicated that patients were mostly overweight/obese. 

### 4.2. MD Scoring

Adherence to the MD was originally measured using a nine-point item scale, which has previously been validated [40] and consisted of a brief questionnaire for assessing adherence to MD and obtaining a rapid feedback. However, to include and control a wider range of food features of MD, adherence to an MD was rescored using the validated 14-point Mediterranean diet score (MDS) based on the PREDIMED score, where a score of 0 indicates lowest adherence to an MD and a score of 14 indicates highest adherence [3]. Recent MD advice was taken into consideration [41] and types and quantities of foods within the PREDIMED score were adapted to reflect the typical diet and dietary recommendations in Northern Ireland [42].

### 4.3. ^1^H-NMR Analysis of Blood 

Plasma samples were collected from 58 participants (n = 137) at baseline (n = 58), 6 months (n = 44) and 12 months (n = 35). After blood collection, samples were left undisturbed at room temperature for 15 minutes for the blood to clot. Samples were centrifuged (2000× *g*; 10 min) and the supernatant (plasma) collected, aliquoted and stored at −80 °C. Plasma filtering was carried out using an adapted version of a previously used method [43]. Prior to filtration, centrifugal filter units (Amicon Ultracel; 0.5 mL; 3 kDa cut-off) were rinsed seven times each with H_2_O (0.5 mL) and centrifuged at 12,000× *g* for 20 min, to remove residual glycerol bound to the filter membranes. Each plasma sample (300 μL) was transferred to the centrifuge filter units. The samples were then centrifuged (13,000× *g* for 30 min at 4 °C) to remove macromolecules, which consist primarily of proteins and lipoproteins. Filtered plasma (285 µL) from each sample was transferred to a separate microcentrifuge tube. To each sample, D_2_O (35 μL) and buffer solution (30 μL; 11.7 mM DSS [disodium-2,2-dimethyl-2-silapentane-5-sulphonate], 1.75 M K_2_HPO_4_ (anhydrous), and 5.84 mM 2-chloro pyrimidine-5-carboxylic acid in H_2_O) was added. Samples (200 μL) were subsequently transferred to a standard Bruker 3mm NMR tube for analysis. Data collection was carried out on a 600 MHz Bruker ASCEND NMR spectrometer equipped with a five mm TCI cryoprobe using randomised order with two hundred and fifty-six transients acquired. Chemical shifts were reported in parts per million (ppm) of the operating frequency. DSS was used as the internal standard for chemical shift referencing and quantification. Using Bayesil (a web-based system that consists on a library of pure compounds specifically designed for the identification and quantification of 1D ^1^H-NMR metabolites) [44,45] all collected spectra were profiled using a custom library of 59 metabolites. 

### 4.4. Statistical Analysis

To explore the metabolite associations, the 14-point MDS (Appendix A) was used to divide the samples into two groups. The highest MDS achieved by any individual on the 14-point scale was 10; hence, the resulting groups were: Low Score: (0–4) and High Score (5–10) by splitting at the median (Appendix A). Results obtained from these two groups were statistically analysed by performing univariate analysis (t-test with the software MetaboAnalyst (version 4.0) [46] to obtain the most significant features from identified metabolites. All data were normalised and auto-scaled (normalisation by sum and auto-scaling on Metaboanalyst) prior to univariate and multivariate analysis. False discovery rates (FDR, *q*-value) were calculated to account for multiple comparisons applying the Bonferroni correction on Metaboanalyst. *p*-value, FDR adjusted *p*-values (*q*-values) and fold-change were calculated. Principal Component Analysis (PCA), Partial Least Squares-Discriminant Analysis (PLSDA) and Orthogonal Projections to Latent Structures Discriminant Analysis (OPLS-DA) were carried out with both MetaboAnalyst (v4.0) and Simca (v14.0; Umetrics, Umea, Sweden) to observe group discrimination. PCA was also used to identify potential outliers, followed by PLS-DA to highlight significant metabolites which explained the maximum amount of variation between groups. An important issue with PLS-DA is deciding on the number of latent variables to be used to build the model. The default option offered by Metaboanalyst was used where the optimal number of latent variables was determined by Q^2^ (cross validated R^2^). Variable importance in projection (VIP) plots was used to identify the most influential metabolites responsible for the observed separation between groups. ROC curve analysis is widely used to describe the trade-off between sensitivity and specificity with regard to biomarker performance. The area under the ROC curve (AUROC) value is a robust measure used for biomarker discovery. MetaboAnalyst was used to develop ROC curves of individual metabolites (Table 1). In order to enhance this performance, pair-wise ratios of all possible metabolite concentrations were computed by Metaboanalyst showing those top-ranked ratios (Table 2). This procedure was applied for biomarker discovery rather than performance evaluation, due to the potential of over fitting. Data were tested for normality of distribution using the Shapiro Wilk test (IBM SPSS Statistics 25). All variables tested for correlations were nonparametric and therefore Spearman’s rank correlation coefficient was obtained. Correlations were deemed significant where *p* was ≤ 0.05. Multiple comparison correction was applied to *p*-values of the metabolite-food groups correlation using the Benjamini-Hochberg approach on SPSS. The effects of smoking on blood metabolites were assessed by conducting a t-test between smoker/non-smoker groups using Metaboanalyst.

## 5. Conclusions

In conclusion, this study reports for the first time that there is a potential association between blood levels of pyruvic acid, mannose and *myo*-inositol with MD consumption, and it corroborates previous associations with citric acid and betaine. It is only the third study to examine the relationship between blood metabolites levels and MD and it demonstrates the power of ^1^H-NMR metabolomics in dietary intervention studies. Further studies should explore the validity of these biomarkers/biomarker ratios. Given the heterogeneous nature of the MDS, the metabolites highlighted here are reasonably discriminative, and there is potential to incorporate these into currently available blood-based biomarkers panels of dietary intake. 

## Figures and Tables

**Figure 1 metabolites-09-00201-f001:**
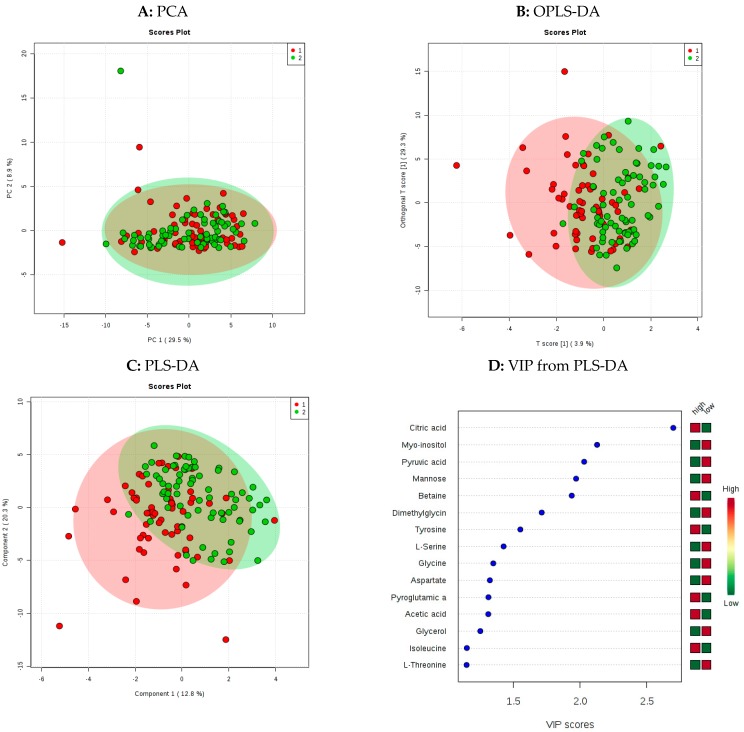
Multivariate statistical modelling of ^1^H-Nuclear Magnetic Resonance ^1^H-NMR metabolomic data. Plots A–C show group separation achieved by principal component analysis (PCA), orthogonal partial least squares discriminant analysis (OPLS-DA) and partial least squares discriminant analysis (PLS-DA). Red circles (●1) represent patients with high MDS and green circles (●2) represent individuals with low MDS. D: Is the resulting variable importance in project (VIP) plot indicating the 15 most influential metabolites responsible for the observed separation in the PLS-DA model.

**Figure 2 metabolites-09-00201-f002:**
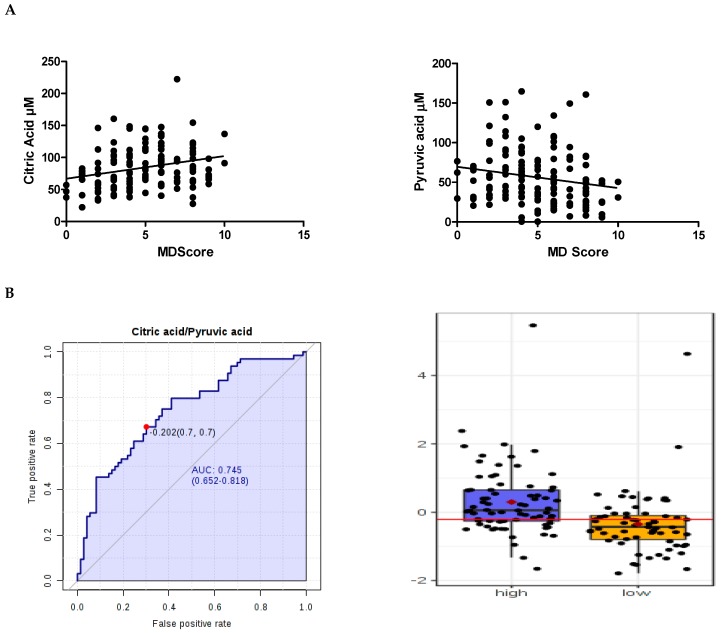
Blood levels of citric acid and pyruvic acid and MDS. (**A**) Citric acid levels positively correlated with MDS, whereas Pyruvic acid levels negatively correlated. (**B**) A paired metabolite ratio of citric acid and pyruvic acid achieved the greatest area under the receiver operating characteristic (ROC) curve (AUC = 0.74) value which was the best performing biomarker for MDS. Optimal cutoff was represented on the curve with a red dot and with a red horizontal line on the box plot. The box-plot showed that the overall distribution profiles for low MDS (0–4) and high MDS (5–10) were broadly similar but the mean value was 53% higher in the high MDS group. Y-axis represents concentrations (µM). Data were median centred. The mean concentration of each group was indicated with a red diamond.

**Table 1 metabolites-09-00201-t001:** Metabolite levels differing according to low or high Mediterranean diet score MDS.

	Name	Mean [µM] Low MDS (n = 64)	SD. Low MDS	Mean [µM] High MDS (n = 73)	SD. High MDS	*p*-Value	*q*-Value (FDR)	AUC (ROC)	↑/↓	% Change
1	citric acid*	76.62	30.50	92.36	32.58	5.99 × 10^−4^*	0.03	0.67	↑	20.5
2	pyruvic acid*	63.50	34.80	50.19	34.36	0.005*	0.16	0.64	↓	−20.9
3	betaine*	40.37	17.83	46.64	18.42	0.01*	0.28	0.62	↑	15.5
4	mannose*	55.08	44.14	39.25	36.58	0.02*	0.32	0.61	↓	−28.7
5	acetic acid	13.06	7.54	14.94	6.55	0.03*	0.34	0.61	↑	14.39
6	*myo*-inositol*	104.67	74.41	79.22	47.25	0.03*	0.34	0.60	↓	−24.3
7	tyrosine	4.58	2.74	5.48	2.86	0.06	0.52	0.59	↑	19.7
8	glycerol	1521.19	817.49	1304.62	875.72	0.08	0.54	0.59	↓	−14.23
9	dimethylglycine	3.01	1.75	2.37	1.56	0.09	0.54	0.58	↓	−21.3
10	malonate	8.35	7.03	8.99	4.35	0.09	0.54	0.58	↑	7.66
11	pyroglutamic acid	87.23	28.57	95.35	29.65	0.10	0.54	0.58	↑	9.3

**Table 2 metabolites-09-00201-t002:** Ratios of paired metabolites improve biomarker utility.

	Name	AUC (ROC)	↑/↓	% Change
1	citric acid/pyruvic acid	0.74	↑	53.3
2	citric acid/l-lactic acid	0.73	↑	29.2
3	citric acid/aspartate	0.72	↑	38.3
4	citric acid/l-phenylalanine	0.71	↑	27.1
5	betaine/pyruvic acid	0.70	↑	47.6

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
