# Peer review of "Application of ^1^H-NMR Metabolomics for the Discovery of Blood Plasma Biomarkers of a Mediterranean Diet"

_metabolites, 2019, doi:10.3390/metabo9100201_

Round 1
Reviewer 1 Report
The study is interesting and valuable. However, I have some questions before the acceptance.
1) To evaluate the metabolites as biomarker perfectly, some additional experiments may be needed. Non-targeted analysis used in this study has some difficulties in quality. Targeted analysis (including sample handling for the metabolites) is needed. If the analysis is difficult, authors should describe quality of the analytical methods for metabolites that authors focused.
2)In some words, Font size is not suitable.
Page 3
A: PCA B: OPLS-DA
Page 9
and even neurodegenerative diseases
However further studies in different populations are necessary to confirm the link between betaine and CVD.
Reviewer 2 Report
The manuscript by Macias et al aims to find associations between the adherence to a Mediterranean diet (MD) and the concentration of metabolites in blood plasma. 1H NMR was used for quantification of plasma metabolites. A 14 point MD scoring system was used for the assessment of the types and quantities of the food. The patients were classified into two groups: high and low, based on the adherence to MD.
The result indicates that the amount of citric acid, pyruvic acid, betaine, mannose, acetic acid and myo-inositol are significantly different in patients with a low compared to a high MD score. These metabolites are also correlated with the consumption of specific food types. I think the paper is well written and that the results are interesting. As already mentioned in the paper, further studies should explore the validity of these metabolites as biomarkers for dietary intake.
I have some questions that should be answered. In addition, some more details should be added to the Methods and Results sections, as specified below.
Comments and questions:
Page 3, line 95: Maybe you can “help the reader” by adding the follow phrase in the first sentence “collected at three different time points (baseline, 6 months, 12 months)”. Ie. Following metabolomic profiling of 137 plasma samples collected at three different time points (baseline, 6 months, 12 months) in 58 subjects, we were able to identify…”
Page 4, Figure 1: In general, it would be easier to read the plot if “High MDS” and “Low MDS” is being used instead of “1” and “2”. Page 4, Fig 1D: According to the figure legend, it looks like “1” (group number 1=red dots) are patients with high MDS. The color scale (to the right) indicates that these patients have low amounts of citric acids, while group 2 (patients with low MDS) have high amounts of citric acid. This result doesn’t fit with the findings observed in the paper. Can you please clarify? A description about the color scale system should be added to the figure legend. In my opinion it would also be easier to read the plot using upwards and downwards arrows instead of color scales.
Page 5, Figure 2: What is the axis in the box plot? Some of the measurements have a value <0. Is the data mean or median centered? Please specify. Please also clarify what the red line in the box plot indicates.
Page 5, line 154: Is the letter “p” missing in the word “box-plot”?
Page 10, line 125: Is the word “samples” missing in the sentence?
Page 11, line 150: Can you please specify in the paper which type of normalization and scaling that was used during multivariate analysis (mean normalization etc)?
Page 11, line 151: Can you please specify which type of correction was used during calculation of FDRs (Bonferroni, Benjamini-Hochberg etc)?
Page 11, line 153: Can you please also indicate how you have chosen the number of latent variables (LVs) in PLS-DA?
Page 11, line 153: Can you please add some information about which type of cross-validation was used for PLS-DA (leave one out etc)?
Many of the measurements in this study are not independent, since up to three measurements from the same patients have been acquired. Are the metabolic profiles from the same patient more similar than that from different patients (given that they follow a similar diet and thus have a similar MD score at the three different time points)? I.e. were the samples from each patients clustered in PCA/PLS-DA. A few sentences should be added to the Discussion section.
Reviewer 3 Report
The first paragraph of the introduction is very long. Please break the material up.
Table 1 needs a lot of work. I'm confused by several things. It presents "Low score", "High score", and "Mean". The use of the word "score" is strange and unclear. "Score" indicates some derived measure that is unitless, but no guidance is given on what "score" might be. The table caption makes me think that I might be looking at concentrations with units of micromolar but it's not clear. Second, why would you report a mean concentration and a standard deviation for all participants pooled together? What would be useful is a table of mean concentration in the 'low MD adherence' group, mean concentration in the 'high MD adherence' group, and standard deviations for both groups.
Table 2: Examining correlations of the top VIP metabolites with MDS scores is perhaps interesting, but I do not follow how the correlations with fruits, vegetables, meat, etc. were performed. How were the food groups quantified? No attempt to correct for multiple comparisons was shown. What is the justification for this calculation? Just because a thing can be calculated doesn't mean it should be calculated, nor should it be used to draw conclusions about data.
Table 3: This is potentially interesting, but the rationale driving the choice of ratios computed is unclear. This needs justification, and I would rather see the authors flesh out this part of the manuscript and omit 90% of Table 2.
Table 4: Again, this is confusing. What data were used to calculate the correlations? I see nothing in the methods about how these food groups were measured or quantified. It must have come from food diaries but the methodology needs to be explained.
Underlying the unclear presentation of the results is a concerning logical conundrum. In the introduction the authors justify the use of metabolomics as arising from a need for an objective biomarker of diet adherence. They cite limitations of food diaries as being subjective. Dietary scores also seem to have limitations in the authors view, but the exact limitations are not well presented. My concern is that the authors created their "high MD adherence" and "low MD adherence" from these scores and then relied heavily upon subjective food diaries for their results. The NMR data are of course objective but the analysis is confounded by the very subjectivity the authors seek to overcome.
Reviewer 4 Report
The manuscript “Application of 1H-NMR metabolomics for the discovery of blood plasma biomarkers of a Mediterranean diet” describes the attempt of discovering plasmatic biomarkers of Mediterranean diet by 1H-NMR metabolomics. The selected strategy seems to be suitable and the topic is really of interest. However, in my view the study fails in the experimental design and the statistics applied. These limitations together with the low sample size (58 participants) make the conclusions extracted questionable. In my view, the limitations of the experimental design should be carefully taken into account in order to obtain reliable results. These aspects should be clarified/addressed/corrected before publication in Metabolites.
Positive Comments:
1.- I agree with the authors about the need of obtaining suitable biomarkers of dietary patterns.
2.- The strategy selected for metabolomics (1H-NMR) is robust .
Comments:
1.- My first concern is related with the experimental design of the study. The authors use samples collected in the frame of the MEDDINI study. All volunteers were diagnosed with CHD and mostly overweight/obese. Both groups have significant differences in blood pressure (both systolic and diastolic) with higher percentage of female and smokers in the low-MDS group. Additionally, the adherence of the participants to the Mediterranean diet was only moderate (only 22 out of 137 samples scored 8-14 in the 14-points scale). I wonder if that is the ideal cohort to identify plasma biomarkers of Mediterranean diet. In my opinion, these limitations must be clearly stated into the text and considered during the statistical analysis (see next point).
2.- As stated before, there are clear differences between the two groups compared in the study regarding blood pressure, sex and percentage of smokers. Therefore, it is not clear if the reported markers are actually markers of dietary patterns or, by contrast, markers of the differences between groups e.g. markers of blood pressure. This result is even more important when some of the proposed markers have been described to be altered by blood pressure e.g. pyruvic and citric. The fact that the differences still persist after correction by these parameters must be evaluated in order to support these metabolites as markers of dietary patterns.
3.-Also regarding with the statistical treatment, several of the 137 samples included in the study belong to the same participant (in fact there are 58 participants in the study). It is not clear the rationale behind using all samples as separate points. I could not find how the fact that 2-3 samples were obtained from the same subject was included into the statistics.
4.- I miss a paragraph reporting also the findings which are not in agreement with previous studies, i.e. those measured metabolites which were proposed as markers of Mediterranean diet which are not confirmed in the present study..
5.- Regarding correlations, r for some Spearman’s tests are very low (e.g. pyruvic vs MD score r = -0.02). Since the p value is statistically significant, these results are treated in the same way that other with larger r values. I miss a more detailed discussion about the meaning of these results.
6.- Table 4: Some of the asterisks are not properly placed or directly missed (e.g. vegetables vs processed meat, fruit vs processed meat, sweet and carbonated drinks vs fruit etc). Please correct. I also wonder if the asterisks are needed taking into account that the actual p value is presented in the table.
7.- Some of the reported metabolites (e.g. pyruvate, citric) belongs to a central metabolic pathway which is also modified by other factors like smoking or exercise. I wonder if these effects were also taken into account.
8.- According to the authors the data was normalized prior to multivariate analysis. Taking into account the limited number of metabolites, it is not clear the method used for normalization.
9.- Reference [20] must be also mentioned in the conclusions section
Round 2
Reviewer 1 Report
I feel this manuscript is acceptable.
Author Response
REVIEWER 1. Comments and Suggestions for Authors
Point1: I feel this manuscript is acceptable.
We thank Reviewer #1 for their appraisal of our manuscript. We are glad they found it interesting and feel it is acceptable.
Reviewer 3 Report
Corrections have significantly improved the clarity of the manuscript. However, I do not understand the relevance of Table 3 to the conclusions drawn. The multitude of weak yet technically statistically significant correlations presented in the manuscript raises concerns about p-hacking. This detracts from the message of the manuscript. The authors should back off of the correlations data.
Author Response
REVIEWER 3. Comments and Suggestions for Authors
Point 1: Corrections have significantly improved the clarity of the manuscript. However, I do not understand the relevance of Table 3 to the conclusions drawn. The multitude of weak yet technically statistically significant correlations presented in the manuscript raises concerns about p-hacking. This detracts from the message of the manuscript. The authors should back off of the correlations data.
We thank Reviewer #3 for their thorough evaluation of our manuscript. We believe their comments and suggestions have added value to our manuscript. We appreciate that corrections have improved the clarity of the manuscript and we thank them for their suggestions to improve it. To comply with the request to ‘back off of the correlations data we have now removed Table 3 from the manuscript and moved it to Supporting data, omitting any information regarding this table in the manuscript in order to avoid detracting from the message of the manuscript.
Reviewer 4 Report
The revised version of the manuscript “Application of 1H-NMR metabolomics for the discovery of blood plasma biomarkers of a Mediterranean diet” has partially solved my main comments. However, in my opinion, the authors fail to properly address/clarify some of the weaknesses of the manuscript. I still consider that these aspects should be clarified/addressed/corrected before publication in Metabolites.
Comments:
1.- The authors have included a list of limitations in the study design. I acknowledge that. However, my main criticisms about the differences between the groups are not commented in the new version of the manuscript:
Since both groups have significant differences in blood pressure (both systolic and diastolic) and some of the proposed markers have been described to be altered by blood pressure e.g. pyruvic and citric acids, I wonder if the reported markers are actually markers of dietary patterns or, by contrast, markers of the differences between groups e.g. markers of blood pressure. Please, clarify.
The adherence of the participants to the Mediterranean diet was only moderate (only 22 out of 137 samples scored 8-14 in the 14-points scale). I wonder if considering a score of 5 out of 14 as high adherence to MD is a valid premise for a scientific paper. Sometimes I have the impression that the authors selected the two groups based only in having two homogenous group without any specific rationale regarding MD adherence.
2.- I could not find in the revised manuscript a mention to the non-controlled parameters which can alter some of the found metabolites such as exercize.
3.- When comparing smokers and non-smokers a p value = 0.07 was obtained for citric acid. Although it does not reach significance, I consider that this trend should be indicated into the text.
Round 3
Reviewer 3 Report
Thank you for moving the table to the supplement. I recommend acceptance for publication.
Author Response
We thank Reviewer #3 for their sound appraisal of our manuscript. We are glad they find it interesting and valuable and that they would like to recommend it for publication.
Reviewer 4 Report
I would like to thank the authors for their comprehensive answers to my comments. I really like the way they answered them and I find that the current version of the manuscript has substantially improved and the results presented are now more solid and reliable for the reader. I only wonder if they could add into the text a brief mention to their comment “An observation from our data was that, when carrying out an ANOVA test between patients at baseline, six months and 12 months, systolic pressure did not significantly change during the intervention (no significant differences between any of the groups, pvalue:0.365). Diastolic pressure showed significant differences between baseline and 12 months (p-value: 4.7E-4) but not between baseline and six months (p-value: 0.138). Interestingly, the highest increase in citric acid was observed between baseline and 6 months, the increase is lower between 6 and 12 months. This follows the same trend as the MD adherence (the increase in scores is higher between baseline and 6 months and lower between 6 months and 12 months). This made us hypothesize that a relationship between citric acid and MD adherence exists.” I find it quite clarifying and I assume that the readers of the manuscript would benefit for having this information. In any case, I consider that the current version of the manuscript deserves publication in Metabolites.
Author Response
REVIEWER 4. Comments for authors.
I would like to thank the authors for their comprehensive answers to my comments. I really like the way they answered them and I find that the current version of the manuscript has substantially improved and the results presented are now more solid and reliable for the reader. I only wonder if they could add into the text a brief mention to their comment “An observation from our data was that, when carrying out an ANOVA test between patients at baseline, six months and 12 months, systolic pressure did not significantly change during the intervention (no significant differences between any of the groups, pvalue:0.365). Diastolic pressure showed significant differences between baseline and 12 months (p-value: 4.7E-4) but not between baseline and six months (p-value: 0.138). Interestingly, the highest increase in citric acid was observed between baseline and 6 months, the increase is lower between 6 and 12 months. This follows the same trend as the MD adherence (the increase in scores is higher between baseline and 6 months and lower between 6 months and 12 months). This made us hypothesize that a relationship between citric acid and MD adherence exists.” I find it quite clarifying and I assume that the readers of the manuscript would benefit for having this information. In any case, I consider that the current version of the manuscript deserves publication in Metabolites.
We thank Reviewer #4 for their in-depth evaluation and sound appraisal of our manuscript. We believe their comments and suggestions have improved the manuscript and have brought more clarity for the reader. We are glad they feel the new version has substantially improved the manuscript and that they recommend this new version for publication.
Following their suggestion, the following paragraph has been added to the manuscript so that the reader can benefit from having this information.
“Systolic blood pressure did not significantly change throughout the intervention when doing an ANOVA test comparing systolic blood pressure at baseline, 6 months and 12 months (p-value: 0.365.). Diastolic blood pressure showed significant differences between baseline and 12 months (p-value: 4.7E-4) but not between baseline and 6 months (p-value: 0.138). The highest increase in citric acid was observed between baseline and 6 months. Citric acid showed a decrease between six and twelve months. Similarly, MD adherence was higher between baseline and 6 and decreased between 6 and 12 months, which strengthens a potential relationship between citric acid and MD adherence.”